# Attitudes of Community Health Nurses Towards Postnatal Home Visitation: A Study in the Ashanti Region of Ghana

**DOI:** 10.3390/ijerph22040534

**Published:** 2025-04-01

**Authors:** Yvonne Agyeman-Duah, Million Bimerew

**Affiliations:** School of Nursing, Faculty of Community and Health Sciences, University of Western Cape, Private Bag X17, Bellville 7535, Western Cape, South Africa; mbimerew@uwc.ac.za

**Keywords:** postnatal home visitation, community health nurse, maternal and child health

## Abstract

Enhancing maternal and newborn care is a key priority for governments worldwide. World leaders have taken deliberate steps to improve these essential services, with an emphasis on postnatal care, including home visits for mothers and their newborns. However, in the Ashanti Region and across Ghana, challenges surrounding the delivery and effectiveness of postnatal home visits remain a significant concern. This study aimed to assess the attitudes of Community Health Nurses towards postnatal home visitation in selected districts of the Ashanti Region. A quantitative survey approach was used to gather data from 100 CHNs randomly selected from 10 regional districts. Structured questionnaires were used to gather quantitative data from CHNs. Findings from the study were analysed using descriptive and inferential tests. The study results indicated that most of the CHNs were young adults, 35 years of age (79%), with the majority having 3–5 years of work experience. The respondents generally demonstrated a negative attitude towards PNHV as they believed it was an extra workload, time-consuming, and unnecessary. A Pearson chi-square test indicated strong significant association between CHNs’ attitude towards the components of the attitude scale and years of experience as well as their age. In conclusion, the study revealed that even though CHNs are expected, as part of their job description, to embark on home visitation activities, most of them have a negative attitude towards this professional duty. There should be conscientious, novel ways to ignite the interest of these essential service providers to help improve infant and maternal health.

## 1. Introduction 

Maternal health is a critical global concern, with significant differences in maternal well-being between high-income and low- to middle-income countries [1]. Despite numerous efforts to enhance maternal and child health through greater utilization of healthcare services, many regions, particularly in Africa and Asia, have not seen the expected progress [2,3].

In low- and lower-middle-income countries, preventable causes and complications related to pregnancy and childbirth contribute to high maternal and neonatal mortality rates. Approximately 800 women die from these causes every day, with 95% of these fatalities occurring in low- and lower-middle-income countries and 70% in sub-Saharan Africa [2,4]. The highest risk of a child’s mortality occurs within the initial 28 days of life. Every year, approximately 2.6 million babies die in their first month of life, with almost half of deaths taking place within the initial 24 h of life, with 75% occurring during the first week [5]. Most infants die at home as a result of infections and complications associated with prematurity, such as respiratory distress syndrome and apnoea [5]. 

The World Health Organization maintains that the postpartum period, the six-week interval between the birth of the newborn and the restoration of the reproductive organs to their pre-pregnancy state, is as strenuous for the mother as it is for the newborn [5,6]. Timely intervention during the postpartum period can potentially avert health issues from progressing into chronic conditions, which could have lasting consequences on women, infants, and their families [7]. The GNSAP emphasizes that in communities with poor access to facilities, home care becomes even more imperative and is practicable with trained community health nurses. Additionally, various cultural and religious beliefs prevent mothers and newborn babies from attending facility-based postnatal care. Some practices require the mother to remain at home until their spouse grants them permission or certain rituals are performed for her and/or the baby [8,9,10]. Skilled care for mothers and newborn survival is critical in the postnatal period [11]. Because of this, the WHO recommendations on Postnatal Care of the Mother and Newborn endorse at least four postnatal visits with mothers and their newborns during the early postpartum period as follows: If the birth is in a health facility, postnatal care in the facility for at least 24 h after birth; if birth is at home, the first postnatal contact should be within 24 h of birth and at least three additional postnatal contacts for all mothers and newborns; on day 3 (48–72 h), between days 7–14 after birth, and six weeks after birth. These visits aim to provide counselling on maternal and infant care, early detection of complications and prompt referral when needed [11].

Community health workers (CHWs) epitomise an extensive approach across the globe to address shortages of health workers and the lack of a ubiquitous national health system, predominantly in rural areas. Providing quality health care services to postnatal mothers in their homes is believed to have a positive influence towards reducing maternal and neonatal mortality. In Ghana, the training of community health nurses is seen as a strategic plan in meeting the health demands of the country, especially in under-resourced communities [12,13]. While many home visitation programs are led by healthcare professionals, volunteer-based models have gained attention for their cost-effectiveness and community-driven approach. Volunteers from the same community understand cultural norms and barriers, making interventions more acceptable and effective [14]. Meta-analyses have shown improvements in health, community support, and parents’ overall self-esteem for those receiving home visiting services [15,16].

The structure and frequency of these visits vary based on national guidelines and local healthcare system capacities. According to Ghana Health Service [17] postnatal care guidelines, CHNs are expected to conduct the following: First visit: Within 24–48 h after delivery (critical period for maternal and neonatal complications). Second visit: Between 3 and 7 days postpartum, focusing on newborn feeding, early danger signs, and maternal recovery. Third visit: Around 14 days postpartum for continued maternal and infant monitoring. Fourth visit: At six weeks postpartum, coinciding with the newborn’s first immunizations and maternal postpartum checkup. Additional visits may be scheduled based on identified risks or complications.

CHNs play a crucial role in providing maternal and child healthcare services, especially in under-resourced areas, by ensuring continuous and accessible care for mothers and newborns [6,18]. Studies have shown that CHNs significantly contribute to improving maternal and neonatal health outcomes through home visits, health education, and early detection of complications [19,20]. CHNs often work in challenging conditions, including high patient loads, insufficient staffing, and a lack of essential resources, which can negatively impact service delivery [21,22]. In many regions, cultural norms and gender-related restrictions hinder CHNs from effectively reaching mothers and newborns for postnatal care [23,24].

The attitudes and behaviours of these maternal and child health care providers (service delivery factors) are critical elements in the success of the home visitation implementation and its quality (process) in the country [25]. It has been suggested that poor working conditions [26], poor reception in the client’s home [27,28], and low staffing [21,29] may elicit negative attitudes in the CHNs. However, there is limited knowledge of the attitudes of CHNs in the Ashanti region of Ghana, hence the need for this study.

## 2. Methods

The researcher used a questionnaire to obtain data on the attitudes of CHNs towards postnatal home visits.

### 2.1. Research Setting

The study was conducted at the postnatal units of district hospitals in the Ashanti Region of Ghana. The Ghana Health Service (GHS), which oversees healthcare delivery in the country, granted permission for data collection in these hospitals.

### 2.2. Sampling and Sample Size

A simple random sampling technique was employed to select 100 Community Health Nurses (CHNs) from a total of 135 CHNs working in 10 district hospitals. The sample size was determined using Krejcie and Morgan’s [30] formula, which provides an appropriate sample size for a given population. To ensure randomness, a list of all eligible CHNs was obtained from the selected hospitals. Each CHN was assigned a unique number, and a random number generator was used to select participants. This process ensured that every CHN had an equal chance of being selected, eliminating selection bias. The number of CHNs selected from each hospital was proportional to the total number of CHNs at that facility. The proportion of CHNs at each hospital was calculated using the formula:Sub-sample size = (x/135) × N
where x represents the number of CHNs in a particular hospital, 135 is the total number of CHNs in all hospitals, and N (100) is the total sample size. This ensured that the sample was representative of the CHN population across the selected hospitals. With a total population of 135 CHNs, selecting 100 CHNs (74%) is a highly representative sample, reducing the risk of sampling error. Krejcie and Morgan’s [30] recommend a sample size of 97 for populations of 135, meaning our chosen 100 CHNs meets the statistical requirements. All the selected CHNs responded to the questionnaire, making the response rate 100%. 

#### Inclusion Criteria

i.CHNs actively involved in maternal and neonatal care who had conducted several postnatal home visits.ii.CHNs who voluntarily consented to participating in the study.

### 2.3. Study Tools

A structured questionnaire was developed to assess CHNs’ attitudes toward postnatal home visits. The questionnaire was adapted from existing literature on nurses’ attitudes [31] and reviewed by experts in midwifery, research methods, and statistics to ensure content validity.

The questionnaire consisted of two sections:

Demographic Information—Included variables such as age, years of experience, and level of education. Attitude Assessment—Comprised 11 items measured on a 4-point Likert scale (ranging from strongly disagree to strongly agree) to evaluate CHNs’ perspectives on postnatal home visits. Domains such as perceived benefits, challenges, professional responsibility, and support systems were included to provide a comprehensive assessment of CHNs’ perspectives. These domains were selected based on their relevance to CHNs’ scope of practice and international maternal health policies from WHO guidelines on postnatal care. To assess the reliability of the questionnaire, a Cronbach’s alpha test was conducted. The results showed reliability coefficients ranging between 0.70 and 0.90, indicating acceptable internal consistency.

### 2.4. Data Collection Process

The researcher gained entry into the selected hospitals after permission had been granted by the GHS and hospital administrators. CHNs who met the inclusion criteria were given the opportunity after they had agreed to participate in the study. Their rights were explained before they were asked to sign the consent form. The questionnaires were administered to the CHNs during their break. It took about 30 min to complete them. Completed questionnaires were returned on the same day of administration.

### 2.5. Ethics

Ethics approval for the study was obtained from the Biomedical Ethics Committee of the University of the Western Cape (BM19/5/8) and the Ethics Review Committee of Ghana Health Services (GHS-ERC002/12/19). Respondents were asked for verbal and written informed consent. The anonymity of respondents was maintained throughout the process of the study by assigning codes to questionnaires. The respondents were informed about the purpose of the study and that their participation was completely voluntary.

### 2.6. Data Analysis

Data were analysed with Statistical Package for Social Sciences (SPSS) version 25. Descriptive statistics were used to describe the general characteristics of respondents in terms of demographics and nurses’ attitudes. Statistical tests such as Chi-square and standard regression were conducted to determine associations between demographic features of CHNs and their attitudes toward postnatal home visitations. The level of significance was set at *p* < 0.05. Data are presented in tables and as narrative descriptions.

## 3. Results

### 3.1. Demographic Characteristics of Community Health Nurses (CHNs)

According to the profile of years of work experience among community health nurses (CHNs) in home visitation as seen in Table 1, approximately one-fifth (20%) of them were in their second year of service. About 77% of CHNs had worked for more than 3 years. The majority of CHNs in the sample were between 20–39 years old (79%), with the largest group being 30–34 years old (41%), followed by 25–29 years (33%).

### 3.2. Attitude of CHNs Towards PNHV

Mixed feelings were noted by community health nurses (CHNs) to indicate levels of positive and negative attitudes towards the postnatal home visitation programme. Data gathered from CHN respondents to assess their attitude towards postnatal home visits are presented in Table 2. 

A few CHNs with a low mean value of (1.32) attested that postnatal home visits are not important to the CHNs. This had a similar import to that of a majority of CHNs who stated that postnatal home visits are not necessary (*M* = 3.53, ±0.611). Some other CHNs, who said mothers should use hospital-based postnatal care only, also claimed that family rather than CHNs should help mothers in postpartum care (*M* = 2.63, ±0.725), declaring that the home environment is not conducive to postnatal care (*M* = 2.63, ±0.774). The notion that postnatal home visits are time-consuming (*M* = 2.56, ±0.880), that postnatal home visits are an extra workload (*M* = 3.04, ±0.909), and that mothers should only use hospital-based postnatal care (*M* = 2.79, ±0.820) did not evince a favourable attitude from most of the CHNs.

### 3.3. Association Between the Age of CHNs and Their Attitude Towards PNHV

The results of the Chi-square test (Table 3) indicated that CHNs’ age is significantly associated with their attitude towards postnatal home visits (*p* < 0.001). Typically, all the CHNs aged 20 years to 39 years disagreed that postnatal home visits are important, and similarly, almost all of them, 93% (n = 93), indicated that postnatal home visits are unnecessary. Still depicting a negative attitude towards postnatal home visits, all CHNs aged 30 years and above indicated that home visits are an extra workload (χ^2^ = 29.08). 

On the positive side, however, most CHNs aged 30 years and above, 54% (n = 54), preferred home visits to working at the hospital. Most CHNs, particularly those aged below 40 years, refuted the position that there is no equipment for postnatal care at home (χ^2^ = 87.34). This group of CHNs declined the perception that it is difficult educating mothers in the home environment (χ^2^ = 81.57), nor did they support the opinion that mothers and their families (χ^2^ = 73.83) provoke CHN caregivers. A further Chi-square test was run to ascertain whether or not CHNs’ years of service are independent of their attitude towards postnatal home visits.

### 3.4. Association Between CHNs’ Years of Service and Their Attitude Towards Postnatal Home Care

As shown in Table 4, the study established a statistical association, significant at *p* < 0.05, between years of service and the attitude of CHNs towards postnatal home visits. Showing some negative attitudes, CHNs who had less than 3 years of professional practice indicated that home visits are not important (χ^2^ = 8.12), and those who had served for more than 3 years also indicated that home visitation is not necessary. Again, most (54%) of CHNs who had worked for more than 3 years avowed that home visits are an extra workload (χ^2^ = 89.33). 

Evincing a positive attitude, most of the CHNs who had worked for more than 3 years preferred home visits to working at the hospital (χ^2^ = 87.95). Of all CHNs who had worked for less than 5 years, 61% declined the position that there is no equipment for postnatal home visits (χ^2^ = 86.73). Similarly, most of the CHNs with less than 5 years of working experience also objected to the claim that it is difficult to educate mothers in the home environment (χ^2^ = 92.44) and added that the perception that mothers provoke CHNs is unfounded (χ^2^ = 101.72).

## 4. Discussion

The study aimed to identify the attitude of CHNs toward PNHV. Attitudes and behaviours of CHNs are critical elements in the success of the home visitation implementation and its quality (process) in the country [25,32]. Eventually, the quality of care provided to mothers and babies will improve.

The study found that younger CHNs (20–39 years) exhibited more negative attitudes toward postnatal home visits, while older CHNs (40+ years) had more positive perceptions. Similar trends have been reported in global maternal healthcare research. Refs. [21,33], reported that younger nurses expressed greater dissatisfaction with community-based healthcare duties, citing poor working conditions, inadequate supervision, and logistical challenges as key deterrents. Ref. [27], similarly found that younger healthcare providers in low-resource settings were more resistant to home visits due to security concerns, limited autonomy, and family resistance. Conversely, older CHNs (40+ years) in the current study demonstrated a stronger preference for home visits, aligning with research by [18], which found that experienced maternal health workers in low-income countries were more committed to community-based postnatal care. Refs. [34,35], reported that more experienced CHWs were more engaged in maternal and neonatal healthcare, as they had a better understanding of community dynamics, stronger problem-solving skills, and a sense of professional fulfilment. Refs. [21,26], emphasized that nurses with longer service experience tend to adopt more positive attitudes toward home visits because they have seen the long-term impact of postnatal care interventions in reducing maternal and neonatal mortality.

Generally, the CHNs demonstrated a negative attitude towards home visitation. Evidently, most of the CHNs did not consider home visitation to be important (*M* = 1.32), seeing it as an extra burden to their already overloaded work (*M* = 3.04) and time-consuming (*M* = 2.56). Clearly, the CHNs are interested in working from their facilities without embarking on any home visitation programme. Even though some of the CHNs preferred home visits to working at the hospitals, a good number of them believed that the home environment is not conducive for postnatal care (*M* = 2.63). They would prefer the mothers to visit the health facilities for postnatal care rather than embarking on home visitation programmes or better still, the family should be made to take care of the postnatal mother, instead of the nurse. This reflects a socio-cultural belief found among rural people in low- and middle-income countries that suggests that family members and neighbours should care for postnatal mothers instead of seeking help from midwives or CHNs due to the influence they have over the postnatal mother [36,37].

Despite the preference for home visits vis-à-vis working at the facility, this poor attitude towards home visitation may be multifaceted. Given the fact that these nurses work mostly in rural areas, poor working environment and poor conditions of service, such as walking to the clients’ homes [26], may be major deterring factors. Human resource capacity of community health nurses continues to be a major challenge confronting nursing in Ghana, as many newly trained nurses prefer to work in towns and hospitals instead of accepting posting to the rural areas. This, therefore, puts extra stress on the few nurses who accept posting to the rural areas, and they may find it difficult to combine home visits with the activities undertaken at the facility [21,29]. This was demonstrated in the responses of the CHNs as they stated that embarking on home visitation was an extra workload. Similar to this finding, ref. [38], also found that community health workers were not always available for postnatal home visitation due to competing tasks at the facilities, serving as a major barrier to general postnatal services. 

One concern inferred from the response of nurses was the issue of a conducive environment for postnatal home care. Previous studies have shown that nurses conducting home visits have reported a lot of violence including verbal abuse, threats, and sexual violence [39,40,41]. Consequently, the care provider may experience anxiety, traumatic stress disorder, and fear of future home visitation. Fearful or threatened workers may be obliged to avoid visits, shorten visits, or refuse to visit homes in communities that are perceived to be unsafe [27,28].

However, the responses that seem to suggest that some of the CHNs see no importance of home visits and that the family should rather care for the postnatal mothers are of much concern. Home visiting is central in preventive health care services, especially among the vulnerable population. In children under five years, it is plausible that nurse home visitation could lead to fewer acute care visits and hospitalisation by providing early recognition of and effective intervention for problems such as jaundice, feeding difficulties, and skin and cord care in the home setting [42,43]. Ref. [44], proposed that mothers who receive regular home visits from nurses have a reduced likelihood to give birth to babies with low birth weight. This, therefore, requires the CHN to fully participate in this core activity. Previous studies have suggested that CHNs in Ghana cover up to only 10% of postnatal services required by newborns born at home within the first 48 h [45]. This seems to confirm that CHNs have a negative attitude towards postnatal home visitation as they see it to be of no or little importance.

## 5. Limitations

The study was conducted only in 10 districts in the Ashanti Region; it may be challenging to apply the findings in other parts of the country. All CHNs who participated in the study were certificate holders, and the findings may not reflect the challenges of other health care workers with higher training and qualifications involved in home visitation.

## 6. Recommendations

Recommendations are made here to CHNs and community health nursing practitioners, the Nursing and Midwifery Council (NMC), and its affiliates.

There should be a proper orientation to the community of deployment before CHNs are made to start work in these communities. They should be well-oriented to the cultural expectations and norms during home visits and home care.

CHNs should endeavour to learn about the culture of the community of deployment, if necessary, and endeavour to respect the cultural beliefs of the mothers whilst working at promoting adherence to evidence-based practices. They should be willing to read about the cultural practices of the communities in available sources of the literature, including history books, from the internet, and from opinion leaders in the community.

The NMC and its affiliates should conduct periodic monitoring and supervision of the work and the conduct of CHNs during home visitations and how well they keep to the ethics of the profession.

## 7. Conclusions

In conclusion, the study has revealed that even though CHNs are expected, as part of their job description, to embark on home visitation activities, most of them have a negative attitude towards this professional duty. This could negatively affect the objective of maternal and newborn care strategies. Relying on current models of care and strategies to reduce maternal and infant mortality, specifically the home visitation strategies, could prove fatal given the study findings. Consequently, new strategies that will enhance CHNs’ participation and interest in postnatal home visitation are inevitable.

## Figures and Tables

**Table 1 ijerph-22-00534-t001:** Demographic characteristics of community health nurses (N = 100).

Item	Age Category	(N)	(N)
Age of CHNs (years)	20–24	5	5
	25–29	33	33
	30–34	41	41
	35–39	16	16
	40–44	2	2
	45–49	1	1
	50+ years	2	2
Years of service (CHNs)	Within 2 years	20	20
	3–5 years	41	41
	6–9 years	36	36
	10 years and above	3	3

**Table 2 ijerph-22-00534-t002:** Descriptive statistics on CHNs’ attitudes towards postnatal home visitation.

Statements	N = 100	Minimum	Maximum	Mean	Std. Deviation
Postnatal home visits are important		1	3	1.32	0.548
Postnatal home visits are not necessary		1	4	3.53	0.611
Postnatal home visits are time-consuming		1	4	2.56	0.880
I prefer home visits to working at the hospital		1	4	2.54	0.758
Home visits are extra workload		1	4	3.04	0.909
Mothers should use hospital-based postnatal care only		1	4	2.79	0.820
Family rather than CHNs must help in postnatal home care		1	4	2.80	0.725
The home environment is not a conducive place for postnatal care		1	4	2.63	0.774
There is no equipment for postnatal care at home		1	4	2.13	0.747
It is difficult educating mothers and family in the home environment		1	4	2.06	0.952
Mothers and their families can provoke CHN caregivers		1	4	2.16	0.762
Valid N (listwise)					

**Table 3 ijerph-22-00534-t003:** Association between CHNs’ age and their attitude towards postnatal home care.

CHNs’ Attitude		Age (in Years) N (%)				Rating	Chi Square	*p* Value
		20–29	30–39	40–49	50+			
Home visitation is important	Agree	0 (0)	0 (0)	2 (2)	2 (2)	Negative attitude	101.61	0.000
	Disagree	38 (38)	57 (57)	1 (1)	0 (0)			
Home visitation is not necessary	Agree	36 (36)	57 (57)	0 (0)	0 (0)	Negative attitude	37.12	0.000
	Disagree	2 (2)	0 (0)	3 (3)	2 (2)			
Postnatal home visits are time-consuming	Agree	0 (0)	49 (49)	3 (3)	2 (2)	Negative attitude	35.34	0.000
	Disagree	38 (38)	8 (8)	0 (0)	0 (0)			
I prefer home visits to working at the hospital	Agree	0 (0)	47 (47)	3 (3)	2 (2)	Positive attitude	37.43	0.000
	Disagree	38 (38)	10 (10)	0 (0)	0 (0)			
Home visits are an extra workload	Agree	19 (19)	57 (57)	3 (3)	2 (2)	Negative attitude	29.08	0.000
	Disagree	19 (19)	0 (0)	0 (0)	0 (0)			
Mothers should use hospital-based postnatal care only	Agree	10 (10)	57 (57)	3 (3)	2 (2)	Negative attitude	29.45	0.000
	Disagree	28 (28)	0 (0)	0 (0)	0 (0)			
Family rather than CHNs must help in postpartum care	Agree	8 (8)	57 (57)	3 (3)	2 (2)	Negative attitude	28.89	0.000
	Disagree	30 (30)	0 (0)	0 (0)	0 (0)			
Home environment is not a conducive place for postnatal care	Agree	1 (1)	57 (57)	3 (3)	2 (2)	Negative attitude	29.90	0.000
	Disagree	37 (37)	0 (0)	0 (0)	0 (0)			
There is no equipment for postnatal care at home	Agree	0 (0)	24 (24)	3 (3)	0 (0)	Positive attitude	87.34	0.000
	Disagree	38 (38)	33 (33)	0 (0)	2 (2)			
It is difficult educating mothers and family in the home environment	Agree	0 (0)	27 (27)	3 (3)	2 (2)	Positive attitude	81.52	0.000
	Disagree	38 (38)	30 (30)	0 (0)	0 (0)			
Mothers and their families can provoke CHN caregivers	Agree	0 (0)	27 (27)	2 (2)	2 (2)	Positive attitude	73.83	0.000
	Disagree	38 (38)	30 (30)	0 (0)	0 (0)			

**Table 4 ijerph-22-00534-t004:** Association between CHNs’ years of service and their attitude towards postnatal home care.

CHNs Attitude		Years of Service N (%)				Rating	Chi-Square	*p*-Value
		Within 2	3–5	6–9	10+			
Home visitation is Important	Agree	0 (0)	0 (0)	1 (1)	3 (3)	Negative attitude	107.18	0.000
	Disagree	20 (20)	41 (41)	35 (35)	0 (0)			
Home visitation is not necessary	Agree	18 (18)	41 (41)	36 (36)	3 (3)	Negative attitude	8.12	0.044
	Disagree	2 (2)	0 (0)	0 (0)	0 (0)			
Postnatal home visits are time-consuming	Agree	0 (0)	15 (15)	36 (36)	3 (3)	Negative attitude	83.75	0.000
	Disagree	20 (20)	26 (26)	0 (0)	0 (0)			
I prefer home visits to working at the hospital	Agree	0 (0)	13 (13)	36 (36)	3 (3)	Positive attitude	87.95	0.000
	Disagree	20 (20)	28 (28)	0 (0)	0 (0)			
Home visits are an extra workload	Agree	1 (1)	41 (41)	36 (36)	3 (3)	Negative attitude	89.33	0.000
	Disagree	19 (19)	0 (0)	0 (0)	0 (0)			
Mothers should use hospital-based postnatal care only	Agree	0 (0)	33 (33)	36 (36)	3 (3)	Positive attitude	93.88	0.000
	Disagree	20 (20)	8 (8)	0 (0)	0 (0)			
Family rather than CHNs must help in postpartum	Agree	0 (0)	31 (31)	36 (36)	3 (3)	Negative attitude	85.65	0.000
	Disagree	20 (20)	10 (10)	0 (0)	0 (0)			
Home environment is not conducive place for postnatal care	Agree	0 (0)	24 (24)	36 (36)	3 (3)	Negative attitude	77.62	0.000
	Disagree	20 (20)	17 (17)	0 (0)	0 (0)			
There is no equipment for postnatal care at home	Agree	0 (0)	0 (0)	26 (26)	3 (3)	Positive attitude	86.73	0.000
	Disagree	20 (20)	41 (41)	10 (10)	0 (0)			
It is difficult educating mothers and family in the home environment	Agree	0 (0)	2 (2)	27 (27)	3 (3)	Positive attitude	92.44	0.000
	Disagree	20 (20)	39 (39)	9 (9)	0 (0)			
Mothers and their families can provoke CHN caregivers	Agree	0 (0)	0 (0)	29 (29)	3 (3)	Positive attitude	101.72	0.000
	Disagree	20 (20)	41 (41)	7 (7)	0 (0)			

## Data Availability

The data presented in this study are available on request from the corresponding author due to ethical restrictions.

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
