# Peer review of "Attitudes of Community Health Nurses Towards Postnatal Home Visitation: A Study in the Ashanti Region of Ghana"

_ijerph, 2025, doi:10.3390/ijerph22040534_

Round 1
Reviewer 1 Report
Comments and Suggestions for Authors
The reviewed article “Attitude of Community Health Nurses’ Towards Post-Natal Home Visitation: A study in the Ashanti region of Ghana” is a paper on the important topic of the attitudes of nurses regarding post-natal visits. The paper needs to address some critical issues before it can be considered for publication.
General comments:
- The sample still generally looks a bit small and more clarification of the sampling and how it is representative might be helpful.
- More literature on what former studies on CHNs should be added.
- You mention “inclusion criteria” – what was it?
- It might be helpful to describe the nature of the visit the CHNs are doing, cause there is some worldwide diversity. When after birth do they visit? How many visits? How long is each visit? What are they expected to do?
- Did you ask how many visits they have conducted during a specific time? Maybe it is associated with perspectives.
- Why are most of the statements in the questionnaire phrased negatively?
- The main aspects tested statistically – age and years of survey findings are not discussed so much in the discussion.
- “new strategies, that will enhance CHNs’ participation and interest in postnatal home visitation is inevitable” – this should be developed - what can be these strategies?
Specific comments:
- No need to mention Statistical Package for Social Sciences (SPSS) in the abstract.
Author Response
COMMENT 1
- The sample still generally looks a bit small and more clarification of the sampling and how it is representative might be helpful.
RESPONSE
- The sample size for the study was enough. The number used was in reference to the population of Community Health Nurses in the selected districts. Refer Page 3, Line 103.
COMMENT 2
- More literature on what former studies on CHNs should be added.
REPONSE
- This observation has been duly addressed. Refer Page 2, Line 77
COMMENT 3
- You mention “inclusion criteria” – what was it?
RESPONSE
- This has been explained in Page 3, Line 121
COMMENT 4
- It might be helpful to describe the nature of the visit the CHNs are doing, cause there is some worldwide diversity. When after birth do they visit? How many visits? How long is each visit? What are they expected to do?
REPONSE
- This has been further explained on Page 2, Lines 69 -73
COMMENT 5
- Did you ask how many visits they have conducted during a specific time? Maybe it is associated with perspectives.
RESPONSE
- The nurses recruited for the study were those who had conducted several postnatal home visits. Specific questions were not asked as stated in the above comment.
- COMMENT 6
- Why are most of the statements in the questionnaire phrased negatively?
RESPONSE
- The authors do agree to this comment
COMMENT 7
- The main aspects tested statistically – age and years of survey findings are not discussed so much in the discussion.
RESPONSE
- This has been explained further on Page 7, Lines 233 - 250
COMMENT 8
- “new strategies, that will enhance CHNs’ participation and interest in postnatal home visitation is inevitable” – this should be developed - what can be these strategies?
RESPONSE
- That was the essence of the entire research. A new implementation strategy has been developed out of several findings from the objectives that were set. However, the strategy is not part of this objective.
Specific comments:
- No need to mention Statistical Package for Social Sciences (SPSS) in the abstract.
Reviewer 2 Report
Comments and Suggestions for Authors
Manuscript Review
The paper addresses a crucial maternal and newborn care issue, specifically focusing on postnatal home visitation (PNHV) programs. However, substantial improvements are necessary for this manuscript to make a meaningful contribution to the readership and the broader research field.
1. Literature Review
The literature review is insufficient in several aspects:
• There is a lack of discussion on successful postnatal home visitation programs worldwide, particularly those based on volunteer models and their advantages.
• A significant portion of the references (nearly half) are outdated (pre-2017). Including more recent studies would also strengthen the discussion and ensure the findings are contextualized within current research trends.
Additionally, home visitation programs vary widely, not only in terms of professional versus volunteer-based models, but also in frequency, content, and scope. The paper does not specify which type of program the nurses were responding to, making it difficult to interpret their perspectives.
2. Methodology
Several methodological concerns need to be addressed:
• The random sampling process is not adequately explained, and it remains unclear how randomness was ensured.
• The sample size (N=100) is relatively small for a quantitative study; a mixed-methods approach incorporating qualitative interviews or focus groups (e.g., with nurses and postpartum mothers) could provide richer insights.
• The questionnaire development is stated to be based on previous literature, but it is unclear how the selected content domains were determined. Further explanation is required to justify why these specific topics were chosen.
Additionally, in the data collection section, the acronym GHS is introduced without explanation. It should be defined when first mentioned.
3. Results Presentation
There are several issues with the presentation of findings:
• The claim that "Some CHNs who showed unsupportive attitudes towards postnatal home visits also claimed that family rather than CHNs should help mothers in postpartum care" is not backed by statistical analysis. The relationship between these variables has not been tested quantitatively, making the assertion unfounded.
• The interpretation of Likert scale mean values (1-4 scale) is problematic. The paper does not clarify what a score of 2.69 versus 2.16 represents, particularly when high standard deviations suggest significant variability in responses.
4. Discussion and Conclusions
The discussion section does not sufficiently engage with the study’s findings, particularly those related to background variables and their associations with CHNs' attitudes (as indicated by Chi-square tests). Consequently, the conclusions and implications lack depth.
Several points require clarification:
• The focus on nurses’ fears regarding home visitation is disproportionate, given that Chi-square tests indicate that most nurses under 40 did not support this concern, and 61% of nurses with less than five years of experience rejected the idea that mothers provoke negative reactions.
• The conclusion section is underdeveloped, failing to consider the study’s key findings and how they could inform policy or practical recommendations.
o For example, how the influence of age and years of experience on nurses' attitudes could be leveraged to assign more supportive nurses to home visits.
o The importance of shaping public perceptions to increase acceptance and demand for PNHV.
Additionally, the lack of discussion on volunteer-based home visitation programs means that alternative solutions to the challenges identified by nurses are not explored, and no recommendation is made to integrate volunteer-based approaches into PNHV programs.
5. Limitations and Contribution to the Field
The study’s limitations further highlight its weak generalizability, and overall, the manuscript does not provide a novel or significant contribution to the existing body of knowledge on postnatal home visitation programs.
Final Recommendation
🔹 Major Revisions Required – The paper requires substantial improvements in its literature review, methodology, data interpretation, discussion, and policy recommendations to contribute meaningful insights to the field.
Author Response
COMMENTS
1. Literature Review
The literature review is insufficient in several aspects:
a. There is a lack of discussion on successful postnatal home visitation programs worldwide, particularly those based on volunteer models and their advantages.
b. A significant portion of the references (nearly half) are outdated (pre-2017). Including more recent studies would also strengthen the discussion and ensure the findings are contextualized within current research trends.
c. Additionally, home visitation programs vary widely, not only in terms of professional versus volunteer-based models, but also in frequency, content, and scope. The paper does not specify which type of program the nurses were responding to, making it difficult to interpret their perspectives.
RESPONSE
The first two comments (a and b) have been addressed. Please refer to Page 2, Lines 76 -94. However, the authors do not agree to comment “c”
Methodology
Several methodological concerns need to be addressed:
a. The random sampling process is not adequately explained, and it remains unclear how randomness was ensured.
b. The sample size (N=100) is relatively small for a quantitative study; a mixed-methods approach incorporating qualitative interviews or focus groups (e.g., with nurses and postpartum mothers) could provide richer insights.
c. The questionnaire development is stated to be based on previous literature, but it is unclear how the selected content domains were determined. Further explanation is required to justify why these specific topics were chosen.
d. Additionally, in the data collection section, the acronym GHS is introduced without explanation. It should be defined when first mentioned.
RESPONSE
All comments under this section has been addressed. Please refer Pages 3 and 4, Lines 107- 149
Results Presentation
There are several issues with the presentation of findings:
a. The claim that "Some CHNs who showed unsupportive attitudes towards postnatal home visits also claimed that family rather than CHNs should help mothers in postpartum care" is not backed by statistical analysis. The relationship between these variables has not been tested quantitatively, making the assertion unfounded.
b. The interpretation of Likert scale mean values (1-4 scale) is problematic. The paper does not clarify what a score of 2.69 versus 2.16 represents, particularly when high standard deviations suggest significant variability in responses.
RESPONSE
The first comment has been addressed. Please refer to Page 5, Lines 186 and 187. The second comment (b) is not understood as the results and interpretations have been clearly stated.
Discussion and Conclusions
The discussion section does not sufficiently engage with the study’s findings, particularly those related to background variables and their associations with CHNs' attitudes (as indicated by Chi-square tests). Consequently, the conclusions and implications lack depth.
Several points require clarification:
a. The focus on nurses’ fears regarding home visitation is disproportionate, given that Chi-square tests indicate that most nurses under 40 did not support this concern, and 61% of nurses with less than five years of experience rejected the idea that mothers provoke negative reactions.
b. The conclusion section is underdeveloped, failing to consider the study’s key findings and how they could inform policy or practical recommendations.
o For example, how the influence of age and years of experience on nurses' attitudes could be leveraged to assign more supportive nurses to home visits.
o The importance of shaping public perceptions to increase acceptance and demand for PNHV.
c. Additionally, the lack of discussion on volunteer-based home visitation programs means that alternative solutions to the challenges identified by nurses are not explored, and no recommendation is made to integrate volunteer-based approaches into PNHV programs.
RESPONSE
All comments raised have been addressed. Please refer to Page 7, Lines 233 -250; And Page 8, Lines 316 -326.
Limitations and Contribution to the Field
The study’s limitations further highlight its weak generalizability, and overall, the manuscript does not provide a novel or significant contribution to the existing body of knowledge on postnatal home visitation programs.
RESPONSE
The authors totally disagree with this assertion. This study findings addresses one out of several objectives set. Mixed method approach was used in gathering data to address all the objectives that were set and a new implementation strategy has been developed out of the findings of the studies conducted.
Reviewer 3 Report
Comments and Suggestions for Authors
The paper gives readers interesting information about a health topic that seems to have a lack of research. The study seems to help fill a gap, so good job in terms of the overall content. I think it would be good to know more about the methods section such as some of the questions that were on the survey. It might be good to share 3-5 questions. Also, did all people contacted take part in the survey? I think there could be more details about the methods.
Author Response
THANKS FOR THE COMMENTS. PLEASE REFER TO PAGES 3 AND 4, LINES 107 TO 149 OF THE EDITED MANUSCRIPT FOR FURTHER DETAILS ON THE METHODOLOGY.
Round 2
Reviewer 1 Report
Comments and Suggestions for Authors
The author made important changes and improved the paper.